# Dual Effects of Beta-Hydroxy-Beta-Methylbutyrate (HMB) on Amino Acid, Energy, and Protein Metabolism in the Liver and Muscles of Rats with Streptozotocin-Induced Type 1 Diabetes

**DOI:** 10.3390/biom10111475

**Published:** 2020-10-23

**Authors:** Milan Holeček, Melita Vodeničarovová, Radana Fingrová

**Affiliations:** Department of Physiology, Faculty of Medicine, Charles University, 500 38 Hradec Králové, Czech Republic; vodenicarovovam@lfhk.cuni.cz (M.V.); rysavar@lfhk.cuni.cz (R.F.)

**Keywords:** branched-chain amino acids, serine, muscles, ATP depletion, ketoglutarate

## Abstract

Beta-hydroxy-beta-methyl butyrate (HMB) is a unique product of leucine catabolism with positive effects on protein balance. We have examined the effects of HMB (200 mg/kg/day via osmotic pump for 7 days) on rats with diabetes induced by streptozotocin (STZ, 100 mg/kg intraperitoneally). STZ induced severe diabetes associated with muscle wasting, decreased ATP in the liver, and increased α-ketoglutarate in muscles. In plasma, liver, and muscles increased branched-chain amino acids (BCAAs; valine, isoleucine, and leucine) and decreased serine. The decreases in mass and protein content of muscles and increases in BCAA concentration were more pronounced in extensor digitorum longus (fast-twitch muscle) than in soleus muscle (slow-twitch muscle). HMB infusion to STZ-treated animals increased glycemia and serine in the liver, decreased BCAAs in plasma and muscles, and decreased ATP in the liver and muscles. The effects of HMB on the weight and protein content of tissues were nonsignificant. We concluded that fast-twitch muscles are more sensitive to STZ than slow-twitch muscles and that HMB administration to STZ-treated rats has dual effects. Adjustments of BCAA concentrations in plasma and muscles and serine in the liver can be considered beneficial, whereas the increased glycemia and decreased ATP concentrations in the liver and muscles are detrimental.

## 1. Introduction

Characteristic features of untreated type 1 diabetes mellitus (T1DM) are hyperglycemia, glycosuria, preferential use of fatty acids as a source of energy, and the loss of body weight associated with reduced muscle mass, muscle weakness, altered function of mitochondria, and overall reduced physical capacity [1,2,3]. Unfortunately, there is accumulating evidence of the limitations of exogenous insulin therapy of T1DM that result in bouts of dysglycemia, dyslipidemia, and insulin resistance, which contribute to the development of long-term diabetic complications, including cardiovascular disease, neuropathy, nephropathy, impaired protein synthesis, and loss of muscle tissue [4,5]. Therefore, there is a strong need for therapeutic approaches to improve diabetes therapies and reduce the incidence of diabetic complications.

Among the substances that can affect skeletal muscles and BCAA metabolism is β-hydroxy-β-methyl butyrate (HMB), a unique metabolite of leucine catabolism. HMB has been shown to decrease proteasome and caspase enzyme activities and reduce the apoptosis of myonuclei [6,7,8]. In humans, positive effects were observed in AIDS- and cancer-related cachexia, pulmonary disease, hip fracture, and other catabolic conditions [9,10,11,12]. Unfortunately, there is little data on the effect of HMB on diabetes. Duan et al. [13] have demonstrated that HMB is more potent than its precursors leucine and ketoisocaproate (KIC) in inhibiting muscle protein degradation, regardless of the presence of insulin. Oral supplementation with a combination of HMB, arginine, and glutamine had beneficial effects on wound healing in diabetic hemodialysis patients [14].

An aspect that has received particular attention in relation to diabetes is the marked increase in plasma concentration of branched-chain amino acids (BCAAs; valine, isoleucine, and leucine), well documented in both T1DM and type 2 diabetes (T2DM) [15,16,17,18]. Recent studies of obesity and subjects with T2DM have suggested that increased BCAA levels play a role in the development of insulin resistance and complications associated with diabetes [18,19,20,21]. However, surprisingly, the pathogenesis of increased BCAA levels has not been clarified in either T1DM or in T2DM.

The aim of our study was to examine how the alterations induced by diabetes might be affected by HMB therapy. We focused on changes in protein and energy balance and amino acid concentrations in plasma, liver, and muscles. We used an animal model of T1DM induced by a single dose of streptozotocin (STZ), which is frequently used for induction of T1DM in rats and mice [22]. Because slow- and fast-twitch fibers exert different sensitivity to various disorders, including diabetes [23,24,25,26,27], musculus soleus (SOL, slow-twitch, red muscle) and musculus extensor digitorum longus (EDL, fast-twitch muscle) were examined.

## 2. Materials and Methods

### 2.1. Animals and Materials

Male Wistar rats (Charles River, Sulzfeld, Germany) weighing approximately 200 g were housed in standardized cages in quarters with controlled temperature and a 12 h light–dark cycle. Rats were maintained on an ST-1 (Velas, Lysa nad Labem, Czech Republic) standard laboratory diet containing (*w*/*w*) 24% nitrogenous compounds, 4% fat, 70% carbohydrates, and 2% minerals and vitamins, and were provided drinking water ad libitum. The Animal Care and Use Committee of Charles University, Faculty of Medicine in Hradec Kralove (license no. 144879/2011-MZE-17214) approved this study on November 1, 2016 (identification code MSMT-33747/2016-4).

Streptozotocin (STZ) was purchased from Sigma-Aldrich (St. Louis, MO, USA), HMB (calcium salt) from Santa Cruz Biotechnology (Dallas, TX, USA), and osmotic pumps from Alzet Osmotic Pumps (Cupertion, CA, USA). Other chemicals were obtained from Sigma-Aldrich (St. Louis, MO, USA), Lachema (Brno, Czech Republic), Waters (Milford, MA, USA), Biomol (Hamburg, Germany), and Merck (Darmstadt, Germany).

### 2.2. Study Design

After 7 days of acclimatization, the rats were randomly assigned to 3 groups. Two groups received a single intraperitoneal injection of STZ (100 mg/kg b.w.) diluted in freshly prepared sterile citrate buffer (0.1 M, pH 4.5), while vehicle was injected into control animals. From day 3 onwards, half of the STZ-treated animals received HMB (200 mg/kg/day) using osmotic pumps placed subcutaneously between the shoulders for 7 days, while vehicle was administered to the remaining animals. Thus, three groups of animals were established: a control group (n = 8), an STZ-treated group (STZ, n = 9), and a group treated with both STZ and HMB (STZ + HMB, n = 8). The dose of STZ was selected based on data from the literature [15,28], and the dose of HMB was selected based on our previous studies [29,30].

At the end of the study, under inhalation anesthesia with 3% isoflurane, the overnight-fasted animals were euthanized by exsanguination from abdominal aorta. The liver, soleus (SOL, slow-twitch, red muscle), and extensor digitorum longus (EDL, fast-twitch, white muscle) muscles were quickly removed and weighed. Small pieces (~0.1 g) of the tissues were immediately frozen in liquid nitrogen. Blood was collected in heparinized tubes and immediately centrifuged for 15 min at 2200× *g* using a refrigerated centrifuge, and the blood plasma was transferred into clean polypropylene tubes. Urine samples were collected from urine excreted during the last 2 h before euthanasia.

### 2.3. Biochemical Markers in Blood Plasma and Urine

Plasma insulin concentration was measured using a rat insulin ELISA kit (Thermo Fisher Scientific, Waltham, MA, USA). Commercial tests purchased from Lachema (Brno, CR) and Sigma-Aldrich (St. Louis, MO, USA) were used for measurement of glucose, urea, creatinine, total protein, and albumin concentrations in blood plasma and/or urine.

### 2.4. Protein Content in Tissues

Protein content in tissues was measured according to Lowry et al. [31]. The results were expressed per gram of wet tissue and per kilogram of body weight.

### 2.5. Amino Acids, Pyruvate, and Branched-Chain Keto Acids (BCKAs) in Blood Plasma and Tissues

Concentrations of amino acids, pyruvate, and BCKAs were determined in supernatants of deproteinized samples of blood plasma and tissues by high-performance liquid chromatography (HPLC) (Aliance 2695, Waters, Milford, MA, USA). Amino acid concentrations were measured after derivatization with 6-aminoquinolyl-N-hydroxysuccinimidyl carbamate using norleucine as an internal standard. o-Phenylenediamine derivatization was used for determination of pyruvate and BCKA (BCKA concentrations in the liver were undetectable). The quinoxalinol derivatives of BCKA were detected using fluorescence with emission and excitation at 410 nm and 350 nm, respectively. The results were expressed per liter of blood plasma or gram of wet tissue.

### 2.6. Adenine Nucleotides

ATP, ADP, and AMP concentrations in the liver and muscles were measured using HPLC. Reversed phase HPLC (Alliance 2695, Waters, Milford, MA, USA) combined with ultraviolet detection was used for the determination of nucleotide concentrations, as described previously [32]. The results were expressed in µmol/g of wet tissue. Energy charge (EC) was calculated according the formula by Atkinson [33]: EC = (ATP + ½ ADP)/(ATP + ADP + AMP).

### 2.7. Tricarboxylic Acid (TCA) Cycle Intermediates in Blood Plasma and Tissues

TCA cycle components, including α-KG, malate, fumarate, oxaloacetate, and cis-aconitate, were quantified by HPLC (Aliance 2695, Waters, Milford, MA, USA) with a YMC-Triart C18 analytical column operated in isocratic mode with 20 mM potassium phosphate buffer (pH 2.9). The wavelength for detection was set at 210 nm. The results were expressed as µmol/g of wet tissue.

### 2.8. Statistical Analysis

The results are expressed as the means ± standard error (SE). Analysis of variance (ANOVA) followed by a Bonferroni multiple comparisons procedure was used to detect differences with a significance level of *p* ˂ 0.05. The statistical software NCSS 2001 (Kaysville, UT, USA) was used for the analyses.

## 3. Results

### 3.1. Biochemical Markers in Blood Plasma and Urine 

All STZ-treated rats became hyperglycemic, with insulin concentrations 10% those of controls (Table 1). STZ treatment led to increased concentrations of urea and ammonia and decreased concentrations of creatinine, total proteins, and albumins in blood plasma. In addition, STZ treatment led to markedly increased excretion of urea and glucose via urine.

In diabetic animals treated with HMB, we found higher plasma concentration of glucose, lower plasma concentrations of albumins, and lower glucose excretion via urine than those observed in diabetic animals without HMB treatment.

### 3.2. Body Weight, Food Intake, and Weights and Protein Contents of Tissues 

There were no differences in animal weight among the experimental groups at the beginning of the study (Table 2). On the last day of the experiment, the body weights of animals with diabetes were lower than those of controls, although food intake was markedly increased in the STZ-treated animals. In STZ-treated animals, the masses and protein contents of the liver and muscles were lower than those of controls. However, the relative values (per kilogram of body weight) of the liver and SOL, but not EDL, were higher in STZ-treated animals than controls.

The effects of HMB on gain of body weight, food intake, and weights and protein contents of tissues of STZ-treated animals were nonsignificant.

### 3.3. Amino Acid Concentrations in Blood Plasma 

Higher concentrations of BCAAs, phenylalanine, ornithine, citrulline, and taurine, and lower concentrations of aspartate, glutamate, glutamine, glycine, and serine were found in blood plasma in both of the STZ-treated groups than in the with controls. In the STZ + HMB group, BCAA and taurine concentrations were significantly lower and glutamine concentration was significantly higher than in the STZ group (Table 3).

### 3.4. Amino Acid Concentrations in Liver 

Increased valine concentrations and decreased concentrations of histidine, asparagine, aspartate, glutamine, glycine, tyrosine, and serine in liver were found in the STZ-treated animals compared with those in the controls. In the STZ + HMB group, higher serine concentration was found when compared with the STZ group without HMB (Table 4).

### 3.5. Amino Acid Concentrations in Muscles 

Marked increases in the concentrations of all three BCAA and phenylalanine were found in both muscles in STZ-treated rats compared with those in control rats (Table 5 and Table 6). The increase in BCAA was more pronounced in EDL (~80%) than in SOL (~50%). In both muscles in STZ-treated rats, decreased concentrations of glutamine, glycine, and serine were found compared with those in control rats. In SOL, alanine, aspartate, and glutamate concentrations were also decreased, whereas in EDL, the proline concentration was increased. HMB therapy attenuated the increase in BCAA concentration induced by STZ treatment in both muscles and decreased aspartate in EDL.

### 3.6. BCKA Concentrations in Blood Plasma and Muscles 

Decreased concentrations of BCKAs in blood plasma were found in STZ-treated animals compared with controls. In EDL, the concentrations of all three BCKAs were increased; in SOL, only α-ketoisocaproate(KIC) was increased, whereas α-ketoisovalerate (KIV) was decreased. The effects of HMB infusion were nonsignificant, with the exception of a decreased KIC concentration in SOL (Table 7). 

### 3.7. Pyruvate and Intermediates of TCA Cycle 

Pyruvate concentration decreased in blood plasma of STZ-treated animals when compared with controls and in EDL of diabetic animals treated by HMB when compared with the STZ group (Table 8). Inconsistent alterations in the concentrations of intermediates of the TCA cycle were observed in STZ-treated animals. Whereas α-KG concentration increased in both muscles, concentrations of other intermediates tended to decrease. The effects of HMB were nonsignificant, with the exception of decreased fumarate concentration in EDL.

### 3.8. Adenine Nucleotide Concentrations in the Liver and Muscles 

In liver, marked decreases in ATP concentration (~50%) resulting in decreased energy charge and increased AMP/ATP were found in STZ-treated animals compared with control animals (Figure 1). The effect of STZ on ATP concentration in muscles was nonsignificant. In diabetic animals treated with HMB, we found lower ATP concentration, lower energy charge, and higher AMP/ATP in all tissues when compared with controls. In SOL of the HMB + STZ group, the ATP concentration was significantly lower than those in control and STZ groups.

## 4. Discussion

To the best of our knowledge, this is the first study to clearly demonstrate alterations in BCAA concentration in different types of skeletal muscle and effects of HMB administration on protein balance, BCAA levels, and energy metabolism in T1DM.

### 4.1. Metabolic Alterations Induced by STZ

Hyperglycemia, insulin concentrations below 10% those of the controls, and glycosuria in all of the STZ-treated rats demonstrated that a single intraperitoneal injection of STZ can be used to create a suitable T1DM model. In addition, STZ induced a severe loss of body weight associated with the loss of skeletal muscle tissue despite hyperphagia. The main cause of impaired muscle protein balance is probably a decrease in protein synthesis, as reported in STZ-treated rats in several studies [34,35]. The marked increases in BCAA concentrations and decreases in serine concentrations in all examined tissues under STZ vs. control treatment (Table 3, Table 4, Table 5 and Table 6 and Figure 2) are consistent with other studies examining aminoacidemia in patients with T1DM and animal models of diabetes [15,16,17,18]. A marked decrease in ATP concentration in the liver and subsequent increase in AMP/ATP may inhibit ATP-consuming processes and stimulate ATP-producing processes via AMP-activated protein kinase (AMPK) activation. These alterations explain the decreased levels of plasma proteins and protein contents in muscles observed in the STZ-treated rats.

### 4.2. Why Are BCAAs Increased in STZ-Treated Animals?

Unlike other amino acids, which are catabolized in the liver, the first site of catabolism of the BCAA is skeletal muscles because of the high activity of BCAA aminotransferase and the skeletal muscle mass. Therefore, the marked increases in BCAA concentrations in the muscles of STZ-treated rats indicated insufficient transamination of the BCAA originating from food and protein breakdown. However, the cause of this insufficient BCAA transamination is not clear.

Because the flux through BCAA aminotransferase is reversible and sensitive to the availability of reactants [36], BCAA transamination is affected by the concentrations of α-KG, BCKAs, and glutamate (Figure 3). Several studies have reported impaired flux through the TCA cycle, increased fatty acid oxidation, and altered function of mitochondria in subjects with T1DM [1,2,3]. These alterations might decrease the availability of α-KG and/or inhibit the flux of BCKA through BCKA dehydrogenase reaction due to increased acyl-CoA availability and an increased NADH to NAD^+^ ratio [37]. Since the muscle concentrations of α-KG increased in STZ-treated rats in our study, the latter possibility, i.e., insufficient utilization of the BCKAs, seems more likely. This speculation is supported by the increased concentrations of all three BCKAs in EDL and of KIC in SOL in the STZ-treated rats. The decrease in BCKA concentration in blood plasma was probably due to an increased activity of hepatic BCKA dehydrogenase, as observed in the liver of subjects with T1DM [38,39].

An increased BCAA concentration in muscles is expected to increase BCAA concentrations in plasma due to decreased BCAA disappearance from the blood and/or increased BCAA release from muscles. A role in pathogenesis of increased plasma BCAA levels might also be played in protein catabolism in the liver. Since the BCAA aminotransferase is absent in the liver, increased catabolism and/or decreased synthesis of hepatic proteins can be expected to result in increased release of the BCAA to the blood [40].

### 4.3. Decreases in Serine Concentration

In our study, the serine concentration was markedly decreased in not only blood plasma (by 51%), but also the liver (67%) and muscles (36% in EDL, 56% in SOL) of STZ-treated rats compared to those in control rats. Decreased plasma serine concentrations have been reported in both T1DM [17,41] and T2DM [42]. It has been suggested that serine depletion plays a role in the development of diabetes mellitus and its related complications, and that L-serine should be considered as an emerging therapeutic option in diabetes [43].

The pathogenesis of serine depletion is unclear. There are two pathways for serine biosynthesis, and both start with 3-phosphoglycerate, an intermediate of glycolysis. Hence, impaired glycolysis due to insulin depletion might play a role in decreased serine concentrations. Glycine can be produced from serine. Hence, decreased synthesis of serine may explain the decreased concentrations of glycine observed in plasma and the examined tissues. 

### 4.4. Effects of HMB 

In diabetic animals treated with HMB, we found higher glucose and insulin concentrations in the blood, lower glucose excretion via the urine, lower ATP concentrations in the liver and muscles, and lower BCAA concentration in blood plasma and muscles than what we observed in STZ-treated rats without HMB treatment. The effects of HMB on the weight and protein content of tissues were nonsignificant (Figure 4).

We do not have a clear explanation for our findings. HMB affects several signaling pathways, including the PI3/Akt and MAPK/ERK pathways, which may interact with and/or replace the effects of insulin and play a role in the pathogenesis of the observed alterations. Some studies have demonstrated that HMB supplementation improves insulin sensitivity [44], whereas others have shown that it impairs insulin sensitivity [45]. Higher glycemia in animals with diabetes treated by HMB when compared with STZ-treated animals without HMB therapy is in favor of studies reporting that HMB decreases sensitivity to the effect of insulin. The suggestion is also supported by significantly higher insulin concentration in HMB-treated rats with diabetes, unless the results were not corrected for multiple testing.

The decrease in BCAA concentrations in plasma and muscles after HMB therapy can be theoretically due to decreased food intake, increased use of the BCAA for protein synthesis, or increased BCAA catabolism. The first two options are unlikely, because we did not observe the effect of HMB on food intake or on the protein balance in examined tissues. However, HMB is likely to activate BCAA oxidation in diabetic rats. It has been suggested that in diabetes, the BCAA increases due to consequences of decreased glycolysis and increased fatty acid oxidation, characteristic features of diabetes, which impair BCAA transamination by decreased supply of amino group acceptors (α-ketoglutarate, oxaloacetate, and pyruvate) and inhibitory influence of NADH and acyl-CoAs on citric cycle and BCKA dehydrogenase [37]. Since it has been proven that HMB promotes the activity of glycolysis-related enzymes [46], it may be supposed that HMB administration improves BCAA transamination in animals with diabetes via improved glycolysis and supply of amino group acceptors. The suggestion that HMB ameliorates increased BCAA levels in diabetes via improved glycolysis is supported by increased hepatic concentration of serine, which is synthesized from 3-phosphoglycerate, an intermediate of glycolysis.

In the light of current opinions about the negative effects of high BCAA concentrations on disease progression and the development of diabetes-associated complications [18,19,20,21], the decreases in BCAA concentration induced by HMB may be considered beneficial. However, it should be noted that there is no consensus in views on the effects of increased BCAA levels on diabetes development [37,47,48]. Attention should be paid to the effects of increased BCAA concentration on protein balance and insulin secretion, and ratio of the BCAA to other amino acids, notably tryptophan and threonine, which may affect serotonin level in the brain and food intake [47].

The decreases in ATP concentrations in the liver and muscles in HMB-treated animals with diabetes should be considered detrimental with potentially adverse influence on the course of diabetes. Decreased ATP concentration and subsequent increase in AMP/ATP might inhibit ATP-consuming processes via AMPK activation, such as protein synthesis. These alterations may explain lower concentrations of albumins in plasma of animals with diabetes treated by HMB when compared with diabetic animals without HMB therapy. Pathogenesis of ATP depletion in liver and muscles of diabetic rats treated by HMB is not clear. One role might be the already mentioned increased drain (cataplerosis) of α-ketoglutarate from TCA cycle due to increased flux of BCAA through BCAA aminotransferase and unknown effects of HMB on mitochondrial function. 

### 4.5. Differences Between SOL and EDL

Our results showed that STZ administration induced greater losses of mass and protein content in EDL than in SOL. This observation is in line with histochemical and morphometric evidence of more severe atrophy of fast-twitch fibers than of slow-twitch ones in rat and murine models of T1DM [23,24]. Another significant difference between SOL and EDL in the response to STZ application was in amino acid concentrations. Under STZ application, BCAA concentration was increased to a greater extent in EDL (by 80%) than in SOL (by 50%). In SOL, but not EDL, the increase in BCAA was associated with marked decreases in alanine, aspartate, and glutamate, amino acids that are closely related to BCAA catabolism (Figure 3).

We assume that the differences between EDL and SOL in response to STZ were due to greater dependence of EDL on glucose as a source of energy, the supply of which to the muscles is impaired under insulin depletion. One role might also be a lower supply of α-KG, an acceptor of amino nitrogen in BCAA transamination reaction produced by TCA cycle, due to a lower content of mitochondria in fast-twitch fibers than in slow-twitch fibers [49]. The finding of decreased alanine, aspartate, and glutamate concentrations in SOL, but not in EDL, under STZ administration might have been due to higher ALT and AST activities in red (slow-twitch) muscles than in white (fast-twitch) muscles [50].

## 5. Conclusions

We conclude that a single dose of STZ induces severe diabetes associated with muscle wasting despite hyperphagia, decreased ATP concentrations in the liver and muscles, and marked increases in BCAA concentrations and decreases in serine concentrations in blood plasma, liver, and muscles. In EDL (white, fast-twitch muscle), the decreases in mass and protein content and the increase in BCAA concentration under STZ treatment were more pronounced than those in SOL (red, slow-twitch muscle). The results suggest that the cause of increased BCAA concentrations following STZ administration is insufficient transamination in muscles. HMB administration to STZ-treated rats had dual roles. The decreases in BCAAs in blood plasma and both muscles and the increase in serine concentration in the liver can be considered beneficial effects of HMB treatment, whereas the higher glycemia and decreased muscle ATP levels should be considered detrimental effects of HMB. Therefore, we cannot conclude whether the effect of HMB on diabetes was beneficial or detrimental.

## Figures and Tables

**Figure 1 biomolecules-10-01475-f001:**
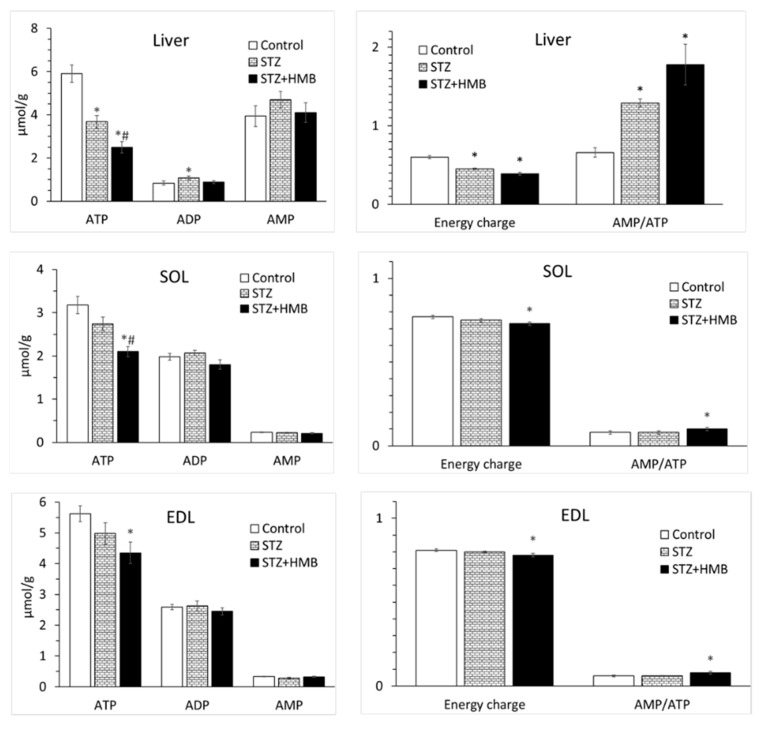
Effect of STZ and HMB on adenine nucleotide concentrations, cellular energy charge, and the AMP-to-ATP ratio in the liver and muscles. Means ± SE, *p* ˂ 0.05. ANOVA and Bonferroni multiple comparisons. * Comparison vs. Control. # Comparison of STZ + HMB vs. STZ. The equation for cell energy charge was: [ATP + (0.5 × ADP)]/[ATP + ADP + AMP].STZ, streptozotocin; HMB, β-hydroxy-β-methyl butyrate. SOL, soleus muscle; EDL, extensor digitorum longus muscle.

**Figure 2 biomolecules-10-01475-f002:**
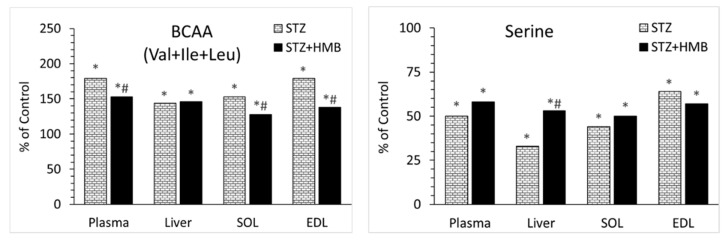
Effects of STZ and HMB on BCAA and serine concentrations in blood plasma, liver, and muscles. Means ± SE, *p* ˂ 0.05. ANOVA and Bonferroni multiple comparisons. * Comparison vs. Control. # Comparison of STZ + HMB vs. STZ.

**Figure 3 biomolecules-10-01475-f003:**
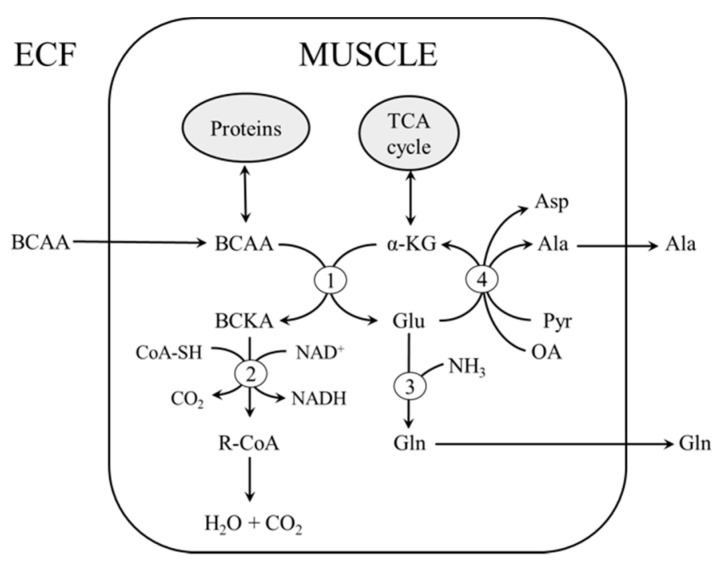
Transamination of the BCAA to BCKA can be affected by supply of the BCAA, activity of TCA cycle, decarboxylation of BCKA, and glutamate conversion to glutamine or α-KG. 1, Branched-chain amino acid aminotransferase; 2, branched-chain α-keto acid dehydrogenase; 3, glutamine synthetase; 4, alanine aminotransferase and aspartate aminotransferase. ECF, extracellular fluid; R-CoA, acyl-CoA.

**Figure 4 biomolecules-10-01475-f004:**
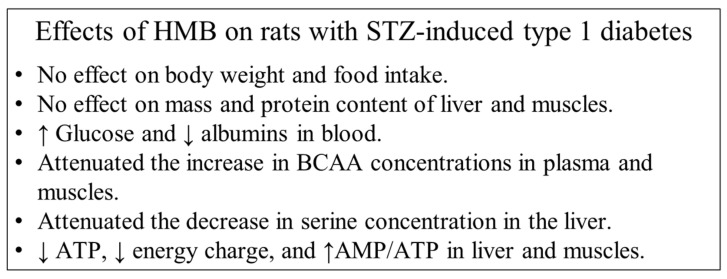
Effects of HMB in rats with STZ-induced type 1 diabetes mellitus.

**Table 1 biomolecules-10-01475-t001:** Biochemical markers in blood plasma and urine.

	Control (n = 8)	STZ (n = 9)	STZ + HMB (n = 8)
Blood plasma			
Glucose (mmol/L)	4.73 ± 0.22	9.37 ± 1.17 *	17.00 ± 2.84 *#
Insulin (µU/mL)	77.53 ± 22.01	3.12 ± 0.46 *	6.25 ± 0.84 *
Urea (mmol/L)	5.37 ± 0.17	18.68 ± 1.00 *	19.63 ± 0.94 *
Ammonia (µmol/L)	35 ± 5	60 ± 6 *	66 ± 6 *
Creatinine (µmol/L)	18.58 ± 0.50	14.67 ± 0.78 *	14.86 ± 0.49 *
Proteins (g/L)	57.65 ± 0.65	51.94 ± 0.85 *	53.56 ± 1.25 *
Albumins (g/L)	38.32 ± 0.70	34.97 ± 0.74 *	29.79 ± 0.61 *#
Urine			
Glucose (mmol/mmol creatinine)	0.35 ± 0.02	590.54 ± 55.80 *	386.23 ± 59.66 *#
Urea (mmol/mmol creatinine)	93 ± 5	319 ± 25 *	389 ± 28 *

Means ± SE, *p* ˂ 0.05. Analysis of variance (ANOVA) and Bonferroni multiple comparisons. * Comparison vs. Control. # Comparison of STZ + HMB vs. STZ. STZ, streptozotocin; HMB, β-hydroxy-β-methyl butyrate.

**Table 2 biomolecules-10-01475-t002:** Body weight, food intake, and weight and protein content of tissues.

	Control (n = 8)	STZ (n = 9)	STZ + HMB (n = 8)
Body weight			
Initial (g)	205 ± 1	208 ± 1	206 ± 1
Final (g)	268 ± 5	189 ± 5 *	175 ± 6 *
Gain (g)	63 ± 5	–23 ± 2 *	–36 ± 4 *
Food intake			
Average (g/day)	24.7 ± 0.4	34.7 ± 0.6 *	32.7 ± 0.9 *
Cumulative (g)	148 ± 3	208 ± 4 *	196 ± 6 *
Liver			
Weight (g)	8.22 ± 0.17	7.11 ± 0.28 *	6.59 ± 0.32 *
(g/kg b.w.)	30.73 ± 0.36	37.49 ± 0.70 *	37.63 ± 0.83 *
Protein (g)	1.33 ± 0.06	1.20 ± 0.05	1.11 ± 0.06 *
(g/kg b.w.)	4.97 ± 0.18	6.34 ± 0.12 *	6.08 ± 0.16 *
SOL			
Weight (mg)	109 ± 3	87 ± 2 *	83 ± 4 *
(mg/kg b.w.)	407 ± 8	461 ± 10 *	475 ± 3 *
Protein (mg)	16.14 ± 0.64	14.22 ± 0.54	14.60 ± 0.78
(mg/kg b.w.)	60.45 ± 2.64	75.42 ± 3.39 *	80.46 ± 2.99 *
EDL			
Weight (mg)	125 ± 4	80 ± 3 *	71 ± 3 *
(mg/kg b.w.)	465 ± 9	420 ± 6 *	408 ± 5 *
Protein (mg)	19.4 ± 0.7	13.2 ± 0.4 *	12.8 ± 1.0 *
(mg/kg b.w.)	72.5 ± 1.6	69.6 ± 1.0	70.0 ± 2.4

Means ± SE, *p* ˂ 0.05. NOVA and Bonferroni multiple comparisons. * Comparison vs. Control. STZ, streptozotocin; HMB, β-hydroxy-β-methyl butyrate.

**Table 3 biomolecules-10-01475-t003:** Amino acid concentrations in plasma.

Plasma	Control (n = 8)	STZ (n = 9)	STZ + HMB (n = 8)
EAA			
Histidine	55 ± 2	57 ± 1	56 ± 2
Isoleucine	105 ± 4	171 ± 7 *	141 ± 7 *#
Leucine	151 ± 5	269 ± 11 *	228 ± 15 *#
Lysine	376 ± 14	333 ± 9	338 ± 26
Methionine	42 ± 1	42 ± 4	38 ± 4
Phenylalanine	68 ± 2	89 ± 1 *	87 ± 4 *
Threonine	228 ± 7	209 ± 10	188 ± 14
Valine	191 ± 6	360 ± 15 *	316 ± 19 *
BCAA	447 ± 14	800 ± 33 *	685 ± 39 *#
Σ EAA	1216 ± 28	1529 ± 42	1394 ± 40 *#
NEAA			
Alanine	368 ± 23	288 ± 21	335 ± 39
Arginine	110 ± 3	124 ± 4	122 ± 6
Asparagine	55 ± 3	51 ± 3	52 ± 3
Aspartate	13 ± 1	1 ± 1*	2 ± 2*
Citrulline	70 ± 3	156 ± 8 *	133 ± 9 *
Glutamate	125 ± 9	62 ± 4 *	56 ± 4 *
Glutamine	688 ± 18	335 ± 13 *	425 ± 24 *#
Glycine	394 ± 12	240 ± 17 *	233 ± 22 *
Ornithine	39 ± 1	48 ± 3 *	60 ± 3 *
Proline	129 ± 6	114 ± 9	143 ± 23
Serine	259 ± 6	129 ± 8 *	149 ± 18 *
Taurine	212 ± 12	429 ± 24 *	310 ± 44 *#
Tyrosine	83 ± 5	70 ± 3	75 ± 9
Σ NEAA	2655 ± 73	2142 ± 86 *	2172 ± 104 *
Σ Amino acids	3882 ± 97	3671 ± 108	3566 ± 125

The values are in µmol/L of plasma. Means ± SE, *p* ˂ 0.05. ANOVA and Bonferroni multiple comparisons. * Comparison vs. Control; # Comparison of STZ + HMB vs. STZ. STZ, streptozotocin; HMB, β-hydroxy-β-methyl butyrate; EAA, essential amino acids; NEAA, non-essential amino acids.

**Table 4 biomolecules-10-01475-t004:** Amino acid concentrations in the liver.

Liver	Control (n = 8)	STZ (n = 9)	STZ + HMB (n = 8)
EAA			
Histidine	637 ± 17	505 ± 55 *	570 ± 29
Isoleucine	180 ± 8	231 ± 21	214 ± 25
Leucine	288 ± 11	389 ± 33	387 ± 47
Lysine	786 ± 56	1051 ± 212	994 ± 114
Methionine	58 ± 2	54 ± 2	67 ± 5 #
Phenylalanine	105 ± 4	108 ± 4	99 ± 5
Threonine	756 ± 68	613 ± 155	1,481 ± 508
Valine	291 ± 12	477 ± 45 *	512 ± 86 *
BCAA	760 ± 31	1098 ± 98 *	1113 ± 157 *
Σ EAA	3025 ± 183	3293 ± 256	4323 ± 722
NEAA			
Alanine	1199 ± 112	827 ± 41	1199 ± 218
Asparagine	249 ± 12	141 ± 27 *	160 ± 20*
Aspartate	1300 ± 56	685 ± 50 *	1014 ± 186
Glutamate	1436 ± 57	1783 ± 434	1763 ± 104
Glutamine	7334 ± 251	3220 ± 428 *	3818 ± 541 *
Glycine	2887 ± 122	2274 ± 241 *	2486 ± 147
Ornithine	338 ± 18	608 ± 89	975 ± 223 *
Proline	256 ± 23	217 ± 15	321 ± 51
Serine	1104 ± 63	365 ± 15 *	581 ± 83 *#
Taurine	3065 ± 273	2808 ± 339	4800 ± 1075
Tyrosine	103 ± 5	84 ± 3 *	92 ± 7
Σ NEAA	19,927 ± 483	13,180 ± 405*	18,398 ± 1243 #
Σ Amino acids	22,953 ± 650	16,474 ± 603*	22,721 ± 1828 #

The values are in nmol/g of tissue. Means ± SE, *p* ˂ 0.05. ANOVA and Bonferroni multiple comparisons. * Comparison vs. Control; # Comparison of STZ + HMB vs. STZ. STZ, streptozotocin; HMB, β-hydroxy-β-methyl butyrate; EAA, essential amino acids; NEAA, non-essential amino acids.

**Table 5 biomolecules-10-01475-t005:** Amino acid concentrations in soleus (SOL) muscle.

SOL	Control (n = 8)	STZ (n = 9)	STZ + HMB (n = 8)
EAA			
Histidine	426 ± 24	504 ± 46	458 ± 31
Isoleucine	96 ± 4	127 ± 12 *	101 ± 6
Leucine	138 ± 5	209 ± 15 *	172 ± 6 *#
Lysine	941 ± 99	1236 ± 104	1167 ± 158
Methionine	47 ± 2	58 ± 6	48 ± 4
Phenylalanine	71 ± 2	110 ± 4 *	102 ± 6 *
Threonine	708 ± 73	515 ± 86	405 ± 84 *
Valine	175 ± 7	290 ± 22 *	249 ± 11 *#
BCAA	409 ± 15	626 ± 48 *	522 ± 20 *
Σ EAA	2602 ± 161	3048 ± 127	2703 ± 197
NEAA			
Alanine	2698 ± 78	1596 ± 166 *	1454 ± 233 *
Asparagine	515 ± 27	526 ± 38	484 ± 28
Aspartate	2526 ± 114	466 ± 81 *	728 ± 252 *
Glutamate	3686 ± 183	2101 ± 170 *	2075 ± 274 *
Glutamine	8832 ± 405	7019 ± 726 *	7523 ± 442
Glycine	2479 ± 135	2107 ± 122	1685 ± 203 *
Ornithine	67 ± 3	66 ± 5	84 ± 7 *#
Proline	288 ± 9	314 ± 31	273 ± 27
Serine	2699 ± 151	1179 ± 92 *	1353 ± 122 *
Taurine	22,948 ± 712	27,506 ± 1356 *	27,586 ± 1262 *
Tyrosine	102 ± 5	100 ± 5	98 ± 13
Σ NEAA	48,969 ± 1577	45,492 ± 1806	45,501 ± 1328
Σ Amino acids	51,571 ± 1692	48,540 ± 1833	48,204 ± 1409

The values are in nmol/g of tissue. Means ± SE, *p* ˂ 0.05. ANOVA and Bonferroni multiple comparisons. * Comparison vs. Control; # Comparison of STZ + HMB vs. STZ. STZ, streptozotocin; HMB, β-hydroxy-β-methyl butyrate; EAA, essential amino acids; NEAA, non-essential amino acids.

**Table 6 biomolecules-10-01475-t006:** Amino acid concentrations in extensor digitorum longus (EDL) muscle.

EDL	Control (n = 8)	STZ (n = 9)	STZ + HMB (n = 8)
EAA			
Histidine	168 ± 4	213 ± 24	163 ± 11
Isoleucine	113 ± 3	184 ± 13 *	147 ± 6
Leucine	151 ± 4	274 ± 17 *	215 ± 14 *#
Lysine	514 ± 43	593 ± 28	462 ± 51
Methionine	52 ± 2	81 ± 5 *	69 ± 5 *
Phenylalanine	80 ± 2	132 ± 5 *	121 ± 8 *
Threonine	415 ± 31	327 ± 19 *	297 ± 19 *
Valine	203 ± 6	378 ± 25 *	300 ± 13 *#
BCAA	467 ± 13	836 ± 54 *	647 ± 27 *#
Σ EAA	1695 ± 74	2181 ± 66 *	1760 ± 82
NEAA			
Alanine	2192 ± 76	2451 ± 101	2186 ± 140
Asparagine	213 ± 10	292 ± 28 *	238 ± 17
Aspartate	490 ± 22	527 ± 42	383 ± 26 #
Glutamate	989 ± 65	886 ± 157	749 ± 90
Glutamine	3744 ± 205	2443 ± 344 *	2632 ± 200 *
Glycine	3693 ± 221	2193 ± 113 *	2022 ± 150 *
Ornithine	37 ± 2	45 ± 3	42 ± 4
Proline	233 ± 10	412 ± 24 *	375 ± 28 *
Serine	956 ± 49	612 ± 85 *	540 ± 44 *
Taurine	16,188 ± 416	20,297 ± 552 *	19,947 ± 793 *
Tyrosine	117 ± 5	126 ± 5	124 ± 6
Σ NEAA	33,491 ± 685	34,842 ± 746	34,352 ± 980
Σ Amino acids	35,186 ± 737	37,024 ± 766	36,111 ± 1047

The values are in nmol/g of tissue. Means ± SE, *p* ˂ 0.05. ANOVA and Bonferroni multiple comparisons. * Comparison vs. Control; # Comparison of STZ + HMB vs. STZ. STZ, streptozotocin; HMB, β-hydroxy-β-methyl butyrate; EAA, essential amino acids; NEAA, non-essential amino acids.

**Table 7 biomolecules-10-01475-t007:** Concentrations of branched-chain keto acids (BCKA.

	Control (n = 8)	STZ (n = 9)	STZ + HMB (n = 8)
Blood plasma			
KIV	14.4 ± 0.9	10.2 ± 0.8 *	9.5 ± 0.8 *
KIC	42.1 ± 3.0	30.3 ± 2.1 *	27.4 ± 2.9 *
KMV	34.2 ± 2.4	26.1 ± 1.8 *	28.9 ± 2.6
∑ BCKA	90.7 ± 6.1	66.6 ± 4.4 *	65.7 ± 5.0 *
SOL			
KIV	10.5 ± 0.8	7.5 ± 0.6 *	7.7 ± 0.8 *
KIC	1.1 ± 0.2	3.5 ± 0.4 *	2.2 ± 0.3 #
KMV	33.4 ± 3.6	33.6 ± 2.2	26.9 ± 3.2
∑ BCKA	44.7 ± 4.1	44.2± 2.6	36.8 ± 3.4
EDL			
KIV	24.1 ± 0.8	29.6 ± 0.9 *	32.5 ± 2.3 *
KIC	2.2 ± 0.3	6.1 ± 1.2 *	3.8 ± 0.5
KMV	29.0 ± 6.9	50.1 ± 3.7 *	50.0 ± 2.1 *
*∑ BCKA*	55.3 ± 7.2	85.7 ± 5.2 *	86.3 ± 3.9 *

The values are in µmol/L of plasma or nmol/g of tissue. Means ± SE, *p* ˂ 0.05. ANOVA and Bonferroni multiple comparisons. * Comparison vs. Control; # Comparison of STZ + HMB vs. STZ. STZ, streptozotocin; HMB, β-hydroxy-β-methyl butyrate; KIV, α-ketoisovalerate; KIC, α-ketoisocaproate; KMV, α-keto-β-methylvalerate.

**Table 8 biomolecules-10-01475-t008:** Concentrations of pyruvate and intermediates of tricarboxylic acid (TCA) cycle.

	Control (n = 8)	STZ (n = 9)	STZ + HMB (n = 8)
Blood plasma			
pyruvate	539 ± 59	379 ± 26 *	464 ± 24
malate	821 ± 112	1179 ± 224	1297 ± 200
α-ketoglutarate	72 ± 10	104 ± 20	115 ± 18
fumarate	10 ± 1	9 ± 1	10 ± 1
cis-aconitate	44 ± 4	27 ± 1 *	33 ± 2 *
Liver			
pyruvate	18.7 ± 1.6	19.8 ± 1.3	22.1 ± 1.8
malate	16.96 ± 1.49	12.13 ± 1.17 *	11.27 ± 1.46 *
α-ketoglutarate	1.81 ± 0.19	2.08 ± 0.13	2.06 ± 0.18
fumarate	0.17 ± 0.01	0.20 ± 0.02	0.22 ± 0.02 *
oxaloacetate	3.90 ± 0.39	3.64 ± 0.75	2.94 ± 0.22
cis-aconitate	0.25 ± 0.01	0.17 ± 0.01 *	0.19 ± 0.01 *
SOL			
pyruvate	56.1 ± 3.9	42.7 ± 5.1	43.6 ± 7.4
malate	3.49 ± 0.17	3.28 ± 0.30	2.95 ± 0.11
α-ketoglutarate	2.42 ± 0.16	3.00 ± 0.15 *	2.48 ± 0.10
fumarate	0.26 ± 0.01	0.11 ± 0.01 *	0.12 ± 0.01 *
oxaloacetate	139 ± 9	110 ± 4 *	109 ± 3 *
cis-aconitate	0.07 ± 0.03	0.02 ± 0.00	0.02 ± 0.00
EDL			
pyruvate	207 ± 16	263 ± 14	156 ± 23#
malate	5.99 ± 1.08	3.41 ± 0.95	2.06 ± 0.46 *
α-ketoglutarate	1.64 ± 0.32	2.85 ± 0.11 *	3.15 ± 0.49 *
fumarate	0.14 ± 0.00	0.13 ± 0.01	0.09 ± 0.01 *#
oxaloacetate	226 ± 7	217 ± 20	219 ± 10
cis-aconitate	0.012 ± 0.002	0.012 ± 0.001	0.007 ± 0.001 *

The values are in µmol/L of blood plasma or µmol/g of tissue. Means ± SE, *p* ˂ 0.05. ANOVA and Bonferroni multiple comparisons. * Comparison vs. Control; # Comparison of STZ + HMB vs. STZ. STZ, streptozotocin; HMB, β-hydroxy-β-methyl butyrate;

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
