# Peer review of "Dual Effects of Beta-Hydroxy-Beta-Methylbutyrate (HMB) on Amino Acid, Energy, and Protein Metabolism in the Liver and Muscles of Rats with Streptozotocin-Induced Type 1 Diabetes"

_biomolecules, 2020, doi:10.3390/biom10111475_

Round 1
Reviewer 1 Report
Holecek et al examined in vivo effects of HMB on STZ-induced type 1 diabetes model rats and reported that HMB showed dual effects (increased blood glucose and decreased BCAA levels). The manuscript was logically well-written, and the experimental design is fair. The reviewer consider their study to be interesting for the readers.
Minor comment:
Legend for Figure 3 need to be described.
Author Response
Response to the Reviewer No. 1
Comment
Holecek et al examined in vivo effects of HMB on STZ-induced type 1 diabetes model rats and reported that HMB showed dual effects (increased blood glucose and decreased BCAA levels). The manuscript was logically well-written, and the experimental design is fair. The reviewer consider their study to be interesting for the readers.
Minor comment:
Legend for Figure 3 need to be described.
Response
Dear reviewer,
Thank you very much for the positive evaluation of my article. The legend to Figure 3 has been added.
Milan Holecek
Reviewer 2 Report
Authors evaluated the effect of HMB in STZ-induced diabetes model. This report was interesting, but the benefit of HMB was unclear. Several issues had better to be addressed.
- The benefit of HMB was unclear. Furthermore blood glucose and insulin were increased by HMB treatment. Dose modification of STZ or HMB might be considered in present model.
- The histology of liver, SOL or EDL should be shown. Morphological changes or presence of inflammation should be assessed.
Author Response
Response to the Reviewer No. 2
Dear reviewer,
thank you very much for reviewing our study. Here are our answers to your comments:
Comment
The benefit of HMB was unclear. Furthermore blood glucose and insulin were increased by HMB treatment. Dose modification of STZ or HMB might be considered in present model.
Response
We discuss the effects of HMB on glucose and insulin concentration in the revised version of the manuscript (page 13, llines 378-84) as follows: “Some studies have demonstrated that HMB supplementation improves insulin sensitivity [44], whereas others have shown that impairs insulin sensitivity [45]. Higher glycemia in animals with diabetes treated by HMB when compared with STZ-treated animals without HMB therapy is in favor of studies reporting that HMB decreases sensitivity to the effect of insulin. The suggestion is supported also by significantly higher insulin concentration in HMB-treated rats with diabetes, unless the results were not corrected for multiple testing.”
The dose of STZ administered in this study induces severe type 1 diabetes and is consistent with the dose administered by other authors (Am. J. Physiol. 1988, 254, E292-300; Biochem. Mol. Med. 1997,61, 87-94). The appropriate dose of HMB was determined in our lab several years ago and no different effects of HMB were observed, even when administered at higher concentrations. The same dose of HMB as given in this study was effective in our previous study (Physiol. Res. 67:741-751, 2018.
Comment
The histology of liver, SOL or EDL should be shown. Morphological changes or presence of inflammation should be assessed.
Response
There are several studies that show remarkable histological changes in the liver and muscles after STZ application ( Clin Ter. 2009;160(4):283-6; JOP:Journal of the pancreas 2006; 7(4):382-9). However, our study has been focused mainly on the effect of HMB on changes in amino acid metabolism and energy and protein balance in diabetic rats. The main effects of HMB were observed in changes in amino acid levels and energy metabolism. We did not observe the effect of HMB on tissue weight and protein content. Therefore, we believe that routine histological examination is unlikely to demonstrate the effect of HMB on the liver or muscles of diabetic rats in our study. Nevertheless, we admit, that targeted and highly specialized histological examination would be interesting. But this is a task for a separate, morphologically oriented study.
Reviewer 3 Report
The authors investigated the effects of treatment with beta-hydroxy-beta-methylbutyrate (HMB) in a rat model of type 1 diabetes. Rats were treated with streptozotocin, which led to insulin deficiency and hyperglycaemia. The rats also developed muscle wasting and changes in amino acid concentrations in plasma and tissues. Particularly fast twitch muscle seemed sensitive to STZ. HMB may increase muscle anabolic processes or reduce muscle breakdown, and the authors therefore tested whether it can protect against muscle wasting in T1DM. This was not the case according to the results.
This study contributed with novel information and should be considered for publication. However, the effects of HMB are unclear. There are still many unanswered questions, and how HMB actually influences whole-body and muscle metabolism is unclear. The authors measured numerous metabolites, but for the reader it can be hard to grasp the meaning of the different findings. Maybe a more detailed illustration or flowchart, or more figures rather than tables could make it easier to grasp the take-home message faster.
- Based on measurements of ATP, AMP and TCA metabolites, the tissues of animals treated with HMB seemed to suffer from energy deficiency? The authors should discuss this in more detail. Could this be due to disturbances in the TCA cycle or mitochondrial respiration? Nutrient uptake? The authors could further consider measuring other metabolic markers (i.e. muscle glycogen? Lactate? pAMPK? Plasma free fatty acids?).
- HMB mice had higher blood glucose levels. HMB treated mice also seemed to have higher insulin (although not when correcting for multiple testing)? It would be useful if the authors could discuss this finding in more detail. Is if for instance possible that HMB could affect insulin sensitivity (briefly mentioned)?
- The authors conclude that HMG has dual effects, and that a reduction in BCAA is positive. Are we sure this is the case? Amino acid metabolism and balance is in very complex. For instance, feeding with BCAA to increase BCAA levels leads to metabolic disease, but this seems to be due to a shift in the relative concentrations of tryptophan and threonine. https://www.nature.com/articles/s42255-019-0059-2. Although this specific example may not be so relevant to the current study, the authors should be careful to conclude based on BCAA concentrations.
- The authors have adjusted for multiple testing, but have a low number of animals per group. There are many (potential) differences that are not mentioned, because they are not significant after adjusting p-values. It is not always “correct” to have a too stringent focus on p-values. The authors could consider to also comment on some of the “almost-significant” results?
Author Response
Response to Reviewer No. 3
Dear reviewer,
Thank you very much for the favourable evaluation of our article and constructive suggestions. Based on your comments, we have rewritten most of the discussion (page 13) into which we tried to integrate your very inspiring comments. Our responses to your specific comments are as follows:
Comment
Based on measurements of ATP, AMP and TCA metabolites, the tissues of animals treated with HMB seemed to suffer from energy deficiency? The authors should discuss this in more detail. Could this be due to disturbances in the TCA cycle or mitochondrial respiration? Nutrient uptake? The authors could further consider measuring other metabolic markers (i.e. muscle glycogen? Lactate? pAMPK? Plasma free fatty acids?).
Response
We have tried to discuss pathogenesis of energy deficiency due to HMB treatment as follows:
“The decreases in ATP concentrations in the liver and muscles in HMB-treated animals with diabetes should be considered detrimental with potentially adverse influence on the course of diabetes. Decreased ATP concentration and subsequent increase in AMP/ATP might via AMPK activation inhibit ATP-consuming processes, such as protein synthesis. These alterations may explain lower concentrations of albumins in plasma of animals with diabetes treated by HMB when compared with diabetic animals without HMB therapy. Pathogenesis of ATP depletion in liver and muscles of diabetic rats treated by HMB is not clear. A role might play already mentioned increased drain (cataplerosis) of α-ketoglutarate from TCA cycle due to increased flux of BCAA through BCAA aminotransferase and unknown effects of HMB on mitochondrial function.” Please see page 13, lines 406-414.
Comment
HMB mice had higher blood glucose levels. HMB treated mice also seemed to have higher insulin (although not when correcting for multiple testing)? It would be useful if the authors could discuss this finding in more detail. Is if for instance possible that HMB could affect insulin sensitivity (briefly mentioned)?
Response
In the Discussion is stated: “Some studies have demonstrated that HMB supplementation improves insulin sensitivity [44], whereas others have shown that impairs insulin sensitivity [45]. Higher glycemia in animals with diabetes treated by HMB when compared with STZ-treated animals without HMB therapy is in favour of studies reporting that HMB decreases sensitivity to the effect of insulin. The suggestion is supported also by significantly higher insulin concentration in HMB-treated rats with diabetes, unless the results were not corrected for multiple testing”. Please see p. 13, lines 380-384.
Comment
The authors conclude that HMG has dual effects, and that a reduction in BCAA is positive. Are we sure this is the case? Amino acid metabolism and balance is in very complex. For instance, feeding with BCAA to increase BCAA levels leads to metabolic disease, but this seems to be due to a shift in the relative concentrations of tryptophan and threonine. https://www.nature.com/articles/s42255-019-0059-2. Although this specific example may not be so relevant to the current study, the authors should be careful to conclude based on BCAA concentrations.
Response
Accepted. The discussion has been modified (page 13, lines 399-405) as follows: “In the light of current opinions about the negative effects of high BCAA concentrations on disease progression and the development of diabetes-associated complications [18-21], the decreases in BCAA concentration induced by HMB may be considered beneficial. However, it should be noted that there is no consensus in views on the effects of increased BCAA levels on diabetes development [37,47,48]. Attention should be paid to the effects of increased BCAA concentration on protein balance and insulin secretion, and ratio of the BCAA to other amino acids, notably tryptophan and threonine, which may affect serotonin level in the brain and food intake [47].”
Comment
The authors have adjusted for multiple testing, but have a low number of animals per group. There are many (potential) differences that are not mentioned, because they are not significant after adjusting p-values. It is not always “correct” to have a too stringent focus on p-values. The authors could consider to also comment on some of the “almost-significant” results?
Response
I have a similar opinion on p-values. Unfortunately, but not many other opponents. For this reason, according to your proposal, I have included only the effect of HMB on glycemia (p.13, lines 382-384) as follows “The suggestion is supported also by significantly higher insulin concentration in HMB-treated rats with diabetes, unless the results were not corrected for multiple testing”.
Round 2
Reviewer 2 Report
Revised manuscript was well-written and well-addressed for the reviewers' comments.